# The Use of Second-Generation Antipsychotics in Patients with Severe Schizophrenia in the Real World: The Role of the Route of Administration and Dosage—A 5-Year Follow-Up

**DOI:** 10.3390/biomedicines11010042

**Published:** 2022-12-24

**Authors:** Juan J. Fernández-Miranda, Silvia Díaz-Fernández, Francisco López-Muñoz

**Affiliations:** 1Servicio de Salud del Principado de Asturias (SESPA), Asturian Mental Health Service Area V-Hospital Universitario de Cabueñes, 33394 Gijón, Spain; 2Asturian Institute on Health Research-Instituto Para la Investigación Sanitaria del Principado de Asturias (ISPA), 33011 Oviedo, Spain; 3Faculty of Health Sciences, Universidad Camilo José Cela, 28692 Madrid, Spain; 4Neuropsychopharmacology Unit, Hospital 12 de Octubre Research Institute, 28041 Madrid, Spain

**Keywords:** schizophrenia, treatment, outcomes, hospitalisation, adherence, suicide attempts, second-generation antipsychotics, long-acting injectable, high doses

## Abstract

To assess the impact of the route of administration and doses of second-generation antipsychotics (SGAs) on treatment adherence, hospital admissions, and suicidal behaviour in patients with severe schizophrenia (Clinical Global Impression–Severity–CGI-S ≥ 5), we implemented an observational 5-year follow-up study. A total of 37.5% of the patients on oral antipsychotics (Aps) and 11.5% of those on long-acting injectables (LAIs) abandoned the treatment (*p* < 0.001). There were no differences in treatment discontinuation between the LAI-AP standard and high-dose groups. A total of 28.1% of the patients on oral Aps had at least one hospitalisation, as well as 13.1% of patients on LAIs (*p* < 0.001). There were fewer hospitalisations of patients on LAIs in the high-dose group (*p* < 0.05). Suicide attempts were recorded for 18% of patients on oral Aps but only for 4.6% of patients on LAIs (*p* < 0.001). No differences were found between the dosage groups on LAIs. Tolerability was good for all Aps and somewhat better for LAIs than oral Aps in terms of side effects (*p* < 0.05). There were no differences between the standard and high-dose groups. More patients discontinued treatment due to side effects in the oral AP group (*p* < 0.01). LAI SGA treatment was more effective than oral AP in terms of adherence and treatment outcomes for managing people with severe schizophrenia. Moreover, significant improvements were found that favour high-dose LAI SGA treatment for some of these patients. This study highlights the need to consider LAI antipsychotics and high-dose strategies for patients with severe schizophrenia.

## 1. Introduction

Antipsychotic medications are an effective maintenance treatment for schizophrenia and are still the first-line approach [1,2,3,4]. However, patients with schizophrenia are often non-adherent to oral antipsychotic medication, even though they need continuous, long-term treatment [5]. Continuing treatment with antipsychotic medication reduces the risk of relapse in patients with schizophrenia by two-thirds [6,7], and discontinuation studies suggest that more than half of those who stop antipsychotic treatment will relapse within one year, thereby increasing the risk of hospital admission by up to four times [6,8,9]. It is well known that the high prevalence of non-adherence among people with schizophrenia, particularly those with greater clinical severity, seems to be associated with relapses and (re)admissions [10,11,12,13,14,15] and a risk of suicide [16,17,18]. Relapses, measured by hospital admissions, are a common criterion for the effectiveness of treatment for people with schizophrenia [10,11,12,13]. Their relationship with the type of pharmacological treatment (mainly antipsychotic medications) has yet to be clearly established [10,14,18,19,20].

It has been suggested that second-generation antipsychotics (SGAs) are more effective than first-generation antipsychotics (FGAs) in treating negative, cognitive, and depressive symptoms in patients with schizophrenia, and they are also better tolerated than FGAs [15,21,22]. In general, the use of SGAs has already been recommended worldwide [1,3,7,9,23,24].

Long-acting injectable antipsychotics (LAI APs) are considered an effective treatment strategy for improving compliance [14,15,25] and reducing relapses and hospital admissions [20,26,27]. An area of interest is whether or not they can indirectly decrease suicide attempts [16,17,18]. In recent years, recommendations have encouraged the use of LAI antipsychotics as an early treatment option and not only if patients with schizophrenia prefer such treatment, if they have a history of poor adherence, if they have suffered from multiple negative outcomes, such as failed oral medications or multiple relapses and hospitalisations, or if they are very sick, difficult-to-treat patients [28,29,30,31]. LAIs have shown that they are more effective at reducing relapses and hospitalisations compared to oral formulations in randomised controlled trials (RCTs) [14,32,33], as well in naturalistic [33,34,35] and mirror image studies [32,33].

However, although the use of LAI SGAs has already been recommended [3,9,23,28], meta-analyses of randomised clinical trials (RCTs) comparing LAI APs to oral APs (both SGAs) have provided contrasting results [32,33,36,37]. LAI SGAs have shown higher effectiveness in terms of preventing relapse and hospitalisation [23,27,34,35,38] in naturalistic studies [20,34,35,36,38,39,40,41,42]. Some naturalistic studies have even reported fewer hospital admissions at high LAI SGA doses in patients with severe and resistant schizophrenia [43].

Finally, another debated aspect is the use of high doses of APs in complex, severely ill, or resistant patients. The Canadian Optimal Use Report review of the value of high-dose SGAs versus the standard dose for treatment-resistant schizophrenia stated that there is a lack of evidence of any advantages [44]. Moreover, the British consensus on high-dose APs is that there is no justification for the use of high-dose antipsychotic medication for relapse prevention in schizophrenia [45]. Nevertheless, to support the rationale behind high-dose therapy, there are some pieces of evidence: an insufficient amount of antipsychotic medication might not have enough of an effect due to patients’ individual differences in pharmacokinetics, as well as pharmacodynamic differences at the effect site in some people [43,46,47].

There are limited real-world data available that focus on the outcomes of antipsychotic treatments in patients who are more severely ill [4,48]. There are few studies on these key aspects of treatments for people with severe schizophrenia in routine clinical practice (oral vs. LAI, standard vs. high-dose treatment, etc.), with existing studies using small samples and short follow-up times. Research is still needed to confirm whether LAI SGA treatments confer advantages over oral ones in those patients with greater clinical severity and impairment in terms of improved adherence and a reduced risk of admission and suicide. This research seeks to focus on these aspects.

Specifically, the aim here is to assess the impact of the route of administration and doses of SGAs on treatment adherence, clinical severity, hospital admissions, and suicidal behaviour in patients with severe schizophrenia. This study explores whether second-generation antipsychotic long-acting injectables and high-dose treatments may benefit schizophrenia patients with a worse clinical course and who are difficult to stabilise.

## 2. Materials and Methods

We implemented a naturalistic, observational, longitudinal (5-year follow-up) study of patients with severe schizophrenia (ICD-10: F-20; Clinical Global Impressions–Severity scale [CGI-S] ≥ 5). The aim was to compare second-generation oral vs. LAI APs and standard vs. high-dose treatments in terms of treatment compliance, clinical severity, hospital admissions, and suicidal behaviour. The study was carried out in Gijón, Spain (Asturian Mental Health Service–Servicio de Salud del Principado de Asturias, attached to the Spanish National Health Service).

Treatment terminations (all causes), hospital admissions for psychiatric causes, and documented suicide attempts were recorded, along with the AP medication prescribed and the regimen types (oral vs. LAI and standard vs. high-dose treatments). High-dose therapy is defined as doses that exceed the approved recommended dose (as stated by the European Medicines Agency or the USA Food and Drug Administration).

Illness severity was measured using the CGI-S scale at the beginning and during every year of follow-up. AP tolerance was monitored yearly through laboratory tests (haematology, biochemistry, and prolactin levels), with weight and any adverse effects being reported at each visit (no specific scales were used). All psychopharmacological and anti-Parkinsonian medications were registered.

The sample included patients who were undergoing antipsychotic treatment in the Healthcare System in Asturias (Spain). They had to be 18 years of age or older, meet the criteria for diagnosing schizophrenia, and have severe symptoms and impairment, with a CGI-S score of ≥5 (*n* = 516). All of them had been treated with the same AP for at least one year prior to the start of the study. The patients were not randomly assigned to a group (oral or LAI AP, standard or high-dose treatment): the clinics treating the patients decided whether to use oral or LAI APs as well as the doses. The initial recruitment involved 600 subjects, but 34 of them refused to participate in the study, and complete data were not available for 50 of them. A total of 256 patients were treated with oral Aps, and 260 were treated with LAIs. Of the patients on LAIs, 165 were on standard doses and 95 were on high doses. Other clinicians besides those treating the patients were part of the study assessing changes in CGI-S. Recruitment was carried out between January 2015 and December 2016, with the follow-up between January 2017 and December 2021. All the patients (or their legal representatives, where appropriate) signed informed consent forms to begin their treatments. Those on higher-than-recommended AP doses were informed, to which they consented.

The sociodemographic and clinical characteristics of the patients studied are shown in Table 1. The mean age was 41.4 (SD: 10.4). A total of 58.1% were men and 41.9% were women (all patients identified themselves as male or female). As the group of patients receiving the clozapine + LAI combination was quite large (*n* = 59), this subgroup was characterised in terms of demographic and clinical data and compared with the other subgroups. In addition, a separate analysis of the follow-up of this subgroup was performed and compared with the other groups. There were mild but significant differences between the oral, LAI, and clozapine + LAI groups in terms of age, the duration of the previous treatment, co-occurrence with substance use disorders (higher in the LAI AP sample; *p* < 0.05), weight, and CGI-S scores (higher in the clozapine + LAI sample; *p* < 0.05). There were differences between the LAI standard and LAI high-dose groups in terms of length of illness, substance use disorder co-occurrence, and CGI-S scores (higher in the high-dose sample; *p* < 0.05). The mean CGI-S score at the start of the study was 5.4 (1.2) in the group treated with oral APs, 5.8 (0.9) in the group on LAIs, and 6.3 (0.8) in the group on the clozapine + LAI combination (*p* < 0.05). CGI-S scores were higher in the LAI high-dose group compared to the LAI standard-dose group (*p* < 0.05).

The medications prescribed are shown in Table 2. Patients on oral antipsychotics received more antidepressants, anxiolytics, and anti-Parkinsonians than those on LAIs. Among the patients on clozapine and LAI (*n* = 56), only 5 of them were on high doses of LAIs.

The average doses for the LAI high-dose group were RLAI = 110.2 (9.1 SD) mg/14 days, PP = 201.9 (11.3 SD) mg eq./28 days, and AM = 710 (108 SD) mg/28 days.

A descriptive and inferential statistical analysis was conducted. The main statistical analyses involved in comparing oral vs. LAI and standard vs. high-dose APs were treatment discontinuation, hospital admissions, suicide attempts, and CGI-S scores. Chi-square χ2 was used for qualitative variables, with the McNemar test specifically used to compare paired proportions (percentage of patients with treatment discontinuation, hospital admissions, suicide attempts, and LAI APs). Student’s t-test was used for paired data for quantitative variables (number of hospital admissions and suicide attempts, CGI-S scores). The confidence interval was established at 95%. The “R Development Core Team” program (version 3.4.1) MASS Package (7.3-45 version) was used for data processing.

The study was carried out in accordance with the World Medical Association ethical principles (Declaration of Helsinki). The study was approved by the Ethical Clinical Research Committee of the Asturian Regional Hospital (Asturian Health Service–Spanish National Health System)—Comité de Ética de Investigación con medicamentos (CEIm), R 1090/2015, Hospital Universitario Central de Asturias N-1, S3.19 (P.I. 88/16).

## 3. Results

After 5 years, the mean CGI-S score for the oral group was 3.6, and it was 3.3 for the LAI and clozapine + LAI groups (*p* < 0.05); the mean CGI-S score for the LAI standard-dose group was 4 (0.9), and it was 3.9 (0.8) for the LAI high-dose group. The reduction in the CGI-S scores at the end of the study was higher for the LAI and clozapine + LAI samples than it was for the oral AP sample (*p* < 0.01), and it was higher for the LAI high-dose group than it was for the standard-dose group (*p* < 0.05).

A total of 37.5% of the patients on oral APs, but only 11.5% of those on LAIs and 10.1% of patients on the clozapine + LAI combination, abandoned the treatment after 5 years (*p* < 0.001). There were no significant differences in treatment discontinuation between the LAI standard and LAI high-dose groups.

A total of 28.1% of the patients treated with oral APs had at least one psychiatric admission to the hospital during the study (4.7% involuntary). By contrast, only 13.1% of the patients on LAIs and 11.4% of those on clozapine + LAI were admitted to the hospital (*p* < 0.001); among these, 1.5% and 1.6% were involuntary (*p* < 0.001). As far as patients on LAIs were concerned, there were fewer hospitalisations in the high-dose group (*p* < 0.05), with no significant differences in involuntary admissions.

Suicide attempts were recorded for 18% of patients on oral APs but for only 4.6% of those on LAIs and for 4.4% of those on clozapine + LAI (*p* < 0.001); these percentages included those who died as a result of suicide. The average number of attempts was also higher for the oral AP group compared to the LAI and lz + LAI samples (0.2 (0.2) vs. 0.09 (0.1) and 0.08 (0.1); *p* < 0.01). No significant differences were found between the LAI groups (standard and high-dose).

There were no significant differences between the clozapine + LAIs group and the LAIs group except for hospitalisations; there were fewer in the clozapine + LAIs group (*p* < 0.05). No significant association was found for gender in treatment compliance, hospitalisations, involuntary admissions, or suicide attempts in the oral, LAI, and clz + LAI groups or between LAI groups. However, there was an association with regard to hospitalisations and involuntary admissions for the standard-dose group, which were higher than those in the high-dose group (*p* < 0.05).

All these treatment outcomes are summarised in Table 3.

The treatment discontinuation, hospital admissions, and suicide attempts of all subgroups are presented in Figure 1. The survival analysis of the different groups is summarised in Figure 2.

Tolerability was good for all APs, although it was somewhat better for LAIs than it was for oral treatments regarding the side effects reported (*p* < 0.05). There were no significant differences in the side effects reported or in the biological parameters between the standard and high-dose LAI groups, nor were there any in terms of treatment discontinuation due to intolerable adverse effects. However, more patients discontinued treatment due to side effects in the oral AP group than in the LAIs group (*p* < 0.01). Weight gain was lower in the LAI group compared to the oral AP group, but with no statistical significance. The clozapine + LAI group reported more adverse effects (*p* < 0.01), especially sedation and anticholinergic effects (*p* < 0.005), and presented more laboratory test alterations (*p* < 0.01), especially haematology and hepatic function alterations (*p* < 0.01), Moreover, the weight gain in them was significantly higher. Nevertheless, none of these meant a higher risk of abandoning treatment (Table 4).

## 4. Discussion

### 4.1. AP Route of Administration Oral vs. LAI

Recent studies comparing oral SGAs (oral APs) against LAIs in severely ill patients are relatively scarce and have the disadvantage of a brief follow-up period and a relatively small sample size [11,12,49,50,51,52]. The few observational studies that have used administrative databases and included a comparison group have also shown the advantages of LAI APs over oral APs [38,42]. Studies that compare periods of treatment with LAI APs versus oral APs for the same patients may better reflect their impact in the real world [12,18,33,53]. Cohort studies recorded mixed results, but most of them have better results for LAI APs than they do for oral APs [54,55,56].

LAI APs are considered an effective strategy for improving treatment adherence [23,27,28,57,58] and for the early detection of nonadherence [13,59,60], although it is necessary to also consider that quite a large number of patients prefer LAI APs to oral APs because they provide a better quality of life [27,28,39,43]. In general terms, LAI APs seem to have recorded the highest rates of treatment adherence in patients with schizophrenia [23,25,32,34,35,38], as revealed in recent bibliometric studies [57,58].

Nevertheless, the frequent monitoring intrinsic to RCTs better encourages adherence than naturalistic routine clinical practice studies [22,36,61], favouring the oral medication group, and may systematically underestimate the advantages of LAI APs [4,13,25,33,38,55,59,61]. Some naturalistic studies even report higher treatment adherence with high-dose LAI SGAs therapy in patients who are severely ill [33,62,63].

In our research, treatment compliance was significantly higher in people on LAIs than in people on oral APs. Considering the high rates of non-compliance among patients with schizophrenia, especially those with greater severity and impairment, and its association with relapses and hospitalisations [12,23,40,56,60,64], the findings here show how strategies for increasing adherence and therefore reducing decompensations and hospital admissions, such as LAI SGA therapy, are more effective than others, such as oral SGA treatment.

Studies on patients with schizophrenia with a duration of at least 12 months have shown that LAI APs are associated with reductions in relapses and dropouts compared to oral APs [34,40,56,60]. These results are not consistent due to the fact that RCTs that have compared oral APs and LAI APs have often failed to show any clear advantages of LAIs over oral APs in terms of relapse and/or risk of hospital admission. A reason for this may be that they are influenced by the biases of the study design [23,36,42,65]. RCTs involve patients with less severe symptoms or fewer comorbidities, excluding many patients treated in routine practice, particularly those with poor adherence. The better performance of LAI APs compared to oral APs in preventing relapses is better shown by naturalistic and mirror-image studies [12,41,56,60]: the risk of re-admission is at least 20–30% lower with them [12,20,23,40]. Although a meta-analysis that included 21 RCTs did not find any significant differences in the outcome measures [65], other, more recent analyses and systematic revisions between patients treated with LAI APs or oral APs have shown advantages that favour LAIs [32,60,66].

Psychiatric hospital admissions (considered to be relapses) were markedly lower in our study for patients on LAIs, taking into account that the study enrolled people with an increased risk of nonadherence and relapse [12,23,40]. It should also be noted that there were fewer involuntary admissions of patients on LAIs.

Although there is little evidence to suggest that AP medications prevent the risk of suicide, the existing evidence favours SGAs the most, particularly clozapine and LAI SGAs [16,17,18,67]. Clozapine and LAI SGAs have managed to attain the highest rates of treatment compliance and hospitalisation prevention in schizophrenia patients [26,27,35,57]. Whether both facts are clearly linked to suicide prevention is an open question [18,67]. Some meta-analyses did not record any significant differences between LAIs and oral APs with respect to all causes of death, specifically to suicide [13,26,68]. By contrast, recent studies have revealed that LAIs were associated with a lower risk of death (including suicide) than oral APs [34,55]. A meta-analysis of 52 RCTs found no significant difference between LAI APs and oral APs with respect to all causes of death, including suicide [68,69]; although, by contrast, a study featuring a nationwide cohort with schizophrenia found that LAI AP use was associated with a 30% lower risk of death (including suicide) compared to oral APs [39].

In our study, suicide attempts decreased significantly among those patients treated with LAIs compared to those on oral treatments. The role LAIs play in preventing suicide is clearly reinforced by our findings.

Finally, the clozapine + LAI combination group may represent particularly problematic drug-resistant and severely ill patients [43,44,45,46,47,48], and due to the availability of more and more atypical drugs in the LAI formulation, their combination with clozapine becomes an interesting therapeutic option worth analysing in terms of effectiveness and safety. Indeed, this subgroup shows fewer hospitalisations than the only-LAI-AP group (which, in turn, drastically reduces hospital admissions compared to the oral AP group). In addition, and despite having a higher level of adverse effects and alterations in blood tests, there were no more dropouts from treatment compared to the patients who were only on LAIs.

### 4.2. AP Dosages Standard vs. High Doses of LAIs

Due to safety concerns over clozapine and a lack of response in a significant number of patients, treatment with a high-dose SGA is often applied [34,46,48,62,70], even though these strategies have not been recommended in most current guidelines on clinical practice [44,45]. Nevertheless, studies of treatment compliance or relapse prevention do not usually test high-dose regimens. Although those guidelines conclude that higher doses of antipsychotics are not more effective than standard doses, there is some evidence that the average doses of SGAs needed to improve the psychopathology in some severely ill patients are significantly higher than those that are needed for those who are less severely ill. The variability of dose-response relationships among SGAs might be related to differences in their pharmacokinetics and differences in affinities for the different receptors involved in psychopathology and responses to antipsychotics [46,47,63].

In this sense, we can consider two main reasons why higher doses of APs might be justified: insufficient antipsychotics might reach the effect site (pharmacokinetic differences) and the nature of the effect site itself (pharmacodynamic differences). There is sufficient evidence to support the first rationale (low drug plasma levels and an insufficient AP blockade of D2 receptors at standard doses). However, regarding the second rationale, there is currently little evidence to support differences in D2-receptor levels or function in those patients who require high doses [43,48,70]. Nevertheless, the fact is that people with low blood levels usually have bad treatment outcomes [71,72].

In our research, hospital admissions were lower in LAI high-dose regimens than they were in standard regimens. There was no difference regarding treatment adherence between both groups. We point out that the main reason for doses increasing was the lack of effectiveness. The results showed a remarkable improvement in both treatment adherence and effectiveness after changing to a high-dose LAI SGA regimen.

Although the likelihood and intensity of most of the adverse effects of APs increase with dosage, some reactions are unpredictable, and others are neither clearly idiosyncratic nor dose-related [29,46,48,72]. While some studies have found that more patients experience extrapyramidal side effects and elevated prolactin in high-dose SGA treatment than with standard doses, others have reported that fewer patients had Parkinsonism on a high-dose SGA and that the rate of withdrawals due to adverse events is also lower in high-dose SGA patients. The high doses in these patients might also have been better tolerated due to pharmacokinetic or pharmacodynamic reasons [46,64,70]. On balance, the data available do not provide conclusive evidence regarding harm, although several studies point to good tolerability and safety [34,47,48,62,71].

In our study, the tolerability of high doses of LAI SGA (of paliperidone palmitate and aripiprazole once monthly, in particular) is very good, with few, only mild side effects reported and no significant differences for standard doses. This seems to be useful for helping to achieve clinical stabilisation through a high adherence to treatment. Low discontinuation rates also support the good patient acceptance of high doses. Moreover, the safety of high doses has been proven in laboratory test results, with no differences for standard doses.

### 4.3. Strengths and Limitations of the Study

Our study provides real-world results based on clinical practice, avoiding the biases of RCTs. Moreover, it has a large sample size and is the first to compare the treatment adherence, effectiveness, and tolerability of oral APs vs. LAI SGAs in a broad sample of severely ill patients with records of treatment discontinuation over a long period. The same can be said about the comparison between standard and high doses in these patients. Most of the patients on high doses can be considered to be treatment-resistant, and LAI high-dose therapy is an effective and more tolerable alternative to clozapine.

However, some limitations need to be addressed. We employed an open-label non-randomised design under pragmatic conditions, with no real control group, which is a limitation that may mean a lower internal validity. All the patients in our study were rated as severely ill by the CGI-S, so the results may not be generalised for populations of people who are not severely ill. Moreover, we have used the CGI-S for measuring a clinically meaningful change in severity, which is a non-specific instrument and a potential limitation. No formal side effect assessment scales have been applied. Moreover, no blood levels were recorded, so we do not know the relationship between blood levels and the clinical response.

The unmeasured factors related to the underlying clinical decision to put a patient on a specific LAI SGA or oral AP or on standard or high doses are still a limitation for all observational studies in this field.

## 5. Conclusions

This study’s findings show how strategies designed to improve adherence and, consequently, treatment outcomes (reduction in illness severity, hospital admissions, and suicide attempts), such as LAI SGA treatment, are clearly more effective than oral APs in managing people with severe schizophrenia. Our results support the idea that the use of LAI SGAs should be considered much more often as a choice for people with severe schizophrenia who are experiencing impairment and who are at a considerable risk of suicide because they confer advantages over oral APs in terms of improved adherence and a lower risk of hospital admissions and suicidal behaviour among patients in real-world settings.

Moreover, significant improvements have been found that favour the use of high-dose LAI SGA treatment strategies for some of these patients. This is true not only in terms of treatment compliance and effectiveness but also in terms of safety: no clinically significant differences were found between high-dose and standard-dose therapies.

Finally, the clozapine + LAI combination becomes an interesting therapeutic option in terms of effectiveness, compliance, and tolerability, especially for those drug-resistant and more severely ill patients.

We stress that it is not possible to reach definitive conclusions because of the study’s observational design, but this study may contribute to highlighting the need to consider using LAI antipsychotic treatments and high-dose strategies for patients with severe schizophrenia, as we consider both to be more adherent and tolerable treatment options in managing severely symptomatic patients.

Nevertheless, longer-term studies of a sufficient methodological quality and sample size are required to provide a more accurate determination of which LAI SGAs are more effective, tolerable, and adherent and whether high-dose strategies with these APs are clinically viable in treating patients with severe schizophrenia.

## Figures and Tables

**Figure 1 biomedicines-11-00042-f001:**
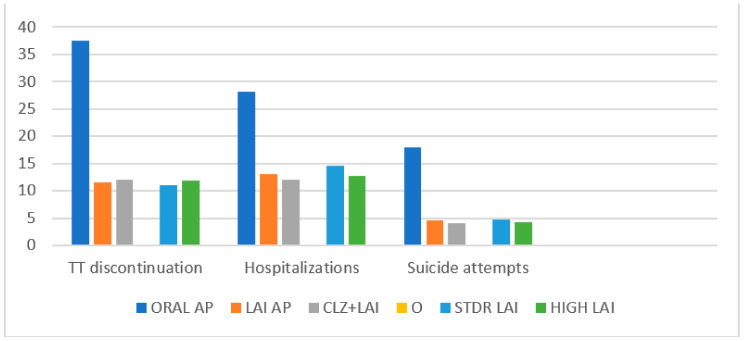
Treatment discontinuation, hospitalisations, and suicide attempts (%).

**Figure 2 biomedicines-11-00042-f002:**
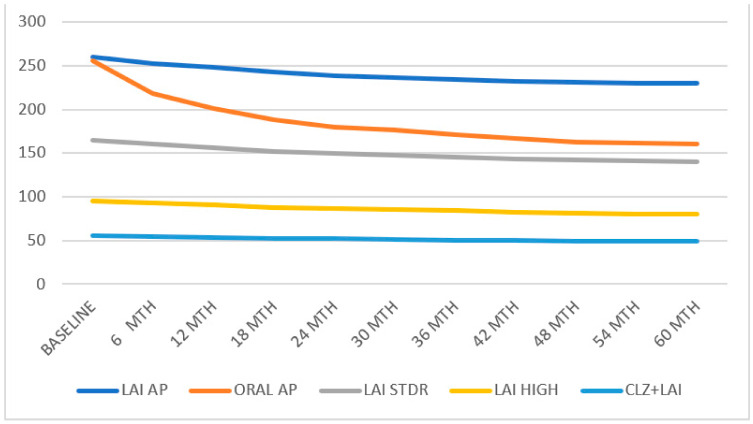
All groups’ treatment discontinuation.

**Table 1 biomedicines-11-00042-t001:** Patients’ sociodemographic and clinical characteristics by the antipsychotic group.

	Total (*n* = 516)	Oral AP (*n* = 256)	LAI AP (*n* = 260)	Clz + LAI (*n* = 56)	LAI Stdr. (*n* = 165)	LAI High (*n* = 95)
Gender: male (*n*, %)	300 (58.1)	162 (63.3)	148 (56.9)	30 (56.8)	98 (59.3)	58 (61.1)
Age [Av (SD)] years	41.4 (10.4)	40.6 (10.8)	43.6 (11.2) *	42.1 (9.6)	40.9 (10.1)	43.1 (10.2)
Length of illness [Av (SD)] years	14.4 (5.7)	13.8 (4.3)	15.2 (5.3)	14.8 (4.9)	14.3 (4.2)	17.8 (5.4) ^ss^
Previous tt duration [Av (SD)] years	10.9 (7)	10 (5.1)	12.1 (8.1) **	11.2 (5.1)	9.6 (5.1)	11 (7.6)
SUD co-occurrence (*n*, %)	182 (35.3)	92 (35.9)	100 (38.5) *	18 (32.1)	58 (35.1)	42 (44.2) ^s^
Somatic diseases co-occurrence (*n*, %)	164 (31.8)	81 (31.6)	83 (31.9)	18 (32.1)	53 (32.1)	30 (31.6)
Weight [Av (SD)] kilograms	75.6 (8.7)	76.5 (9.8)	74.7 (8.1)	77.1 (7.6) *	74.2 (7.8)	75.1 (8.4)
CGI-S scores baseline	5.7 (1.1)	5.4 (1.2)	5.8 (0.9)	6.3 (0.8) *	5.4 (0.6)	6.1 (1) ^s^

(*n*, %): number, percentage; Av (SD): average (standard deviation); AP: antipsychotic; LAI: long-acting injectable; tt: treatment; Clz: clozapine; SUD: substance use disorder; Stdr.: standard. * *p* < 0.05 ** *p* < 0.01 (between oral, LAI, and Clz + LAI groups); ^s^
*p* < 0.05, ^ss^
*p* < 0.01 (between LAI standard and LAI high-dose groups)

**Table 2 biomedicines-11-00042-t002:** Antipsychotics and other medications prescribed.

Antipsychotics Prescribed [N, Av (SD) Doses]	
Oral antipsychotics (*n* = 256) 66 clozapine (578.6 [110.4] mg) 32 risperidone (7.9 [1.3] mg) 23 olanzapine (26.1 [7.2] mg) 10 amisulpride (1020.8 [107.5] mg) 74 aripiprazole (32.9 [9.4] mg) 16 quetiapine (1120.2 [60.1] mg) 9 ziprasidone (214.1 [10.8] mg) 21 paliperidone (12.3 [2.2] mg) 5 others With two or more antipsychotics (*n* = 58)	LAI antipsychotics (*n* = 260) 56 LAI risperidone (56.2 [6.3] mg/14 d) 112 LAI paliperidone (221.3 [30.1] mg/28 d) 88 LAI aripiprazol (706.9 [101.7] mg/28 d) 4 Other LAIs: 4 zuclopenthixol (440 [60.3] mg/14 d) 0 olanzapine With two or more antipsychotics (*n* = 68) (*n* = 56: clozapine + LAI)
Other medications prescribed (oral vs. LAI AP group). Number of patients	
Oral antipsychotics (*n* = 256) Antidepressants: 127 * Mood stabilisers: 28 Anxiolytics: 114 * Anti-Parkinsonians: 118 *	LAI antipsychotics (*n* = 260) Antidepressants: 99 Mood stabilisers: 26 Anxiolytics: 82 Anti-Parkinsonians: 86

(*n*, %): number, percentage; Av (SD): average (standard deviation), tt: treatment. * *p* < 0.01.

**Table 3 biomedicines-11-00042-t003:** Treatment discontinuation, hospital admissions, suicide attempts [*n*, %], and CGI-S scores [Av (SD)] after 5 years.

AP Subgroup (*n* = 516)	Oral AP (*n* = 256)	LAI AP (*n* = 260)	CLZ + LAI (*n* = 56)	LAI–Stdr (*n* = 165)	LAI–High (*n* = 95)
Treatment discontinuation	96 (37.5) **	30 (11.5)	7 (10.1)	19 (11.5)	11 (11.6)
Hospital admissions	72 (28.1) *	34 (13.1)	8 (11.4)	24 (14.5)	10 (10.2) ^s^
Involuntary admissions	12 (4.7) **	4 (1.5)	1 (1.6)	3 (1.7)	1 (1.2) ^s^
Suicide attempts	46 (18) **	12 (4.6)	4 (4.4)	8 (4.9)	4 (4.2)
CGI-S scores	3.6 (1.2)	3.3 (0.9) *	3.3 (0.5) *	3.4 (0.6)	3.2 (0.7) ^s^

(*n*,%): number, percentage of patients; Av: average; SD: standard deviation; AP: antipsychotic; LAI: long-acting injectable; CLZ: clozapine; stdr: standard. * *p* < 0.01, ** *p* < 0.001 (between oral, LAI, and Clz + LAI groups); ^s^
*p* < 0.05 (between LAI–stdr and LAI–high groups).

**Table 4 biomedicines-11-00042-t004:** Adverse effects reported, laboratory tests, and weight [*n* (%)].

Antipsychotic Groups (*n* = 516)	Oral (*n* = 256)	LAI (*n* = 260)	CLZ + LAI (*n* = 56)	LAI–Stdr (*n* = 165)	LAI–High (*n* = 95)
Any adverse effect reported	96 (37.5) *	78 (30)	28 (50) **	51 (30.9)	29 (30.5)
EPS/Parkinsonism	48 (18.7) *	42 (16.1)	8 (11.4) **	26 (15.7)	16 (16.8)
Akathisia	20 (7.8)	18 (7.2)	2 (2.2)	11 (6.7)	7 (7.4)
Sedation	92 (35.8) **	38(14.6)	21 (39.9) ***	24 (14.5)	14 (14.7)
Anticholinergic effects	46 (17.9) **	35 (13.5)	24 (37.2) ***	22 (13.3)	13 (13.7)
Any laboratory test alt. (>20%)	48 (18.7)	50 (19.2)	22 (39.6) **	32 (19.4)	18 (18.9)
Blood count altered	18 (7.2)	9 (3.6)	11 (13.8) **	6 (3.4)	3 (3.2)
Hyperprolactinaemia	40 (15.6) **	27 (10.8) *	4 (5.8)	17 (11.1)	10 (10.2)
Hyperglycaemia	36 (12.5)	35 (13.5)	8 (11.4)	21 (12.7)	14 (14.7)
Hyperlipemia	40 (15.6)	39 (15)	11 (19.8) **	24 (14.5)	15 (15.8)
Hepatic function altered	24 (9.4)	26 (10)	6 (9.3)	16 (9.7)	10 (10.5)
Weight gain (> 10%)	24 (9.4) *	19 (7.3)	11 (19.8) **	16 (9.7)	9 (9.5)
Hypertension	18 (7.2) **	13 (5) *	1 (1.6)	8 (4.9)	5 (5.1)
Tt. discont. due to AE/test alt.	46 (17.9) **	35 (13.5) *	7 (10.1)	16 (9.6)	8 (8.4)

* *p* < 0.05, ** *p* < 0.01, *** *p* < 0.005. AE: adverse effect; alt.: alteration; Tt. discont: treatment discontinuation.

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
