# Peer review of "The Use of Second-Generation Antipsychotics in Patients with Severe Schizophrenia in the Real World: The Role of the Route of Administration and Dosage—A 5-Year Follow-Up"

_biomedicines, 2022, doi:10.3390/biomedicines11010042_

Round 1
Reviewer 1 Report
The article entitled: "Use of second-generation antipsychotics in patients with severe 2 schizophrenia in the real world: role of the route of administration and dosage. A 5-year follow-up" is generally written very well and presents very important study whose results are very valuable for everyday clinical practice and our knowledge of the use and effectiveness of antipsychotics. I have two main comments and the resulting request for supplementation: Firstly, in the description "Patients' sociodemographic, clinical, and AP treatment characteristics" there is no data on: co-occurring somatic diseases, co-occurrence of addictions or substance abuse, and body weight or BMI. If the authors collected this data (or at least some of the data mentioned by me), they should be included in the descriptions of the group and comparisons of subgroups. Secondly, as can be seen, the group of patients receiving the clozapine + LAI combination is quite large. This group should be characterized in terms of demographic and clinical data and compared with other subgroups. In addition, a separate analysis of the follow-up of this subgroup should be performed and compared with groups receiving only oral drugs and only LAIs. This is very important because this group may represent particularly problematic drug-resistant patients and because due to the availability of more and more atypical drugs in the LAI formulation, their combination with clozapine becomes an interesting therapeutic option worth analyzing in terms of effectiveness and safety.
Author Response
|
REVIEWER 1 |

Reviewer 2 Report
The current study compared the treatment outcomes of oral and LAI antipsychotics in a large sample of schizophrenia patients. The results showed that patients on LAI had lower treatment discontinuation rate, fewer hospitalizations, and less suicidal attempts. Those on higher dosage of LAI did not have significant more side effects, and the treatment outcome was as good as standard dose group. The study results provided significant support in clinical treatment of LAI in severe schizophrenia patients. But there were some points need to be clarified.
1. The abbreviations should be carefully reviewed and add the full contents to readers at the first presentence.
2. Some errors in formatting should be corrected. In page 2, line 68 and line 73-75, and page 3, line 112.
3. The format of table 1 should be revised. There were three tables in table 1. It was not clear in “Other treatments (oral vs. LAI AP)” column. Please revised the column, too.
4. The format of table 2 should be revised, too. The clinical characteristics (separated in table 1 and 2) in the oral antipsychotics and LAI group should be summarized in one table. The clinical characteristics of the standard and high dose group could be summarized in one table, too.
5. It would be better if the authors could add survival analysis and a figure regarding treatment discontinuation between oral and LAI treatment group. The hospitalizations in two group could also be presented as figures to show potential difference along the time course.
6. Please provide the definition of standard and high dose group in the methods.
7. Please provide more details of reported side effects and laboratory test alteration in table 3 or in the results.
8. In addition to the CGI-S, was there any other measurements to evaluate the severity of psychotic symptoms in schizophrenia patients, such as PANSS?
Author Response
|
REVIEWER 2 We appreciate reviewer # 2 ´s suggestions and recommendations to improve manuscript quality. |
||
|
The abbreviations should be carefully reviewed and add the full contents to readers at the first presentence. |
Abbreviations had been reviewed and we have added the full contents at the first presentence. |
|
|
Some errors in formatting should be corrected. In page 2, line 68 and line 73-75, and page 3, line 112. |
Errors in formatting on pages 2 and 3 have been corrected. |
|
|
Th The format of table 1 should be revised. There were three tables in table 1. |
The format of table 1 has been changed. Characteristics of all subgroups have been summarized in Table 1a. |
|
|
I It was not clear in “Other treatments (oral vs. LAI AP)” column. Please revised the column, too. |
The medications prescribed are shown in Table 1b. “Other treatments” column has been changed. |
|
|
T The format of table 2 should be revised, too. The clinical characteristics (separated in table 1 and 2) in the oral antipsychotics and LAI group should be summarized in one table. The clinical characteristics of the standard and high dose group could be summarized in one table, too. |
The format of table 2 has been changed. The outcomes of all subgroups have been summarized in one table (Table 2). |
|
|
It would be better if the authors could add survival analysis and a figure regarding treatment discontinuation between oral and LAI treatment group. |
We have added a figure regarding treatment discontinuation through the follow-up (Figure 2). |
|
|
The hospitalizations in two group could also be presented as figures to show potential difference along the time course. |
The treatment abandons, hospitalizations and suicide attempts of all subgroups are presented as figures (Figure 1) |
|
|
P Please provide the definition of standard and high dose group in the methods. |
Definitions are provided in Methods. |
|
|
Please provide more details of reported side effects and laboratory test alteration in table 3 or in the results. |
More details of reported side effects and laboratory test alterations are reported in Table 3. |
|
|
I In addition to the CGI-S, was there any other measurements to evaluate the severity of psychotic symptoms in schizophrenia patients, such as PANSS? |
REVIEWER 2 We appreciate reviewer # 2 ´s suggestions and recommendations to improve manuscript quality. |
|
|
The abbreviations should be carefully reviewed and add the full contents to readers at the first presentence. |
Abbreviations had been reviewed and we have added the full contents at the first presentence. |
|
|
Some errors in formatting should be corrected. In page 2, line 68 and line 73-75, and page 3, line 112. |
Errors in formatting on pages 2 and 3 have been corrected. |
|
|
Th The format of table 1 should be revised. There were three tables in table 1. |
The format of table 1 has been changed. Characteristics of all subgroups have been summarized in Table 1a. |
|
|
I It was not clear in “Other treatments (oral vs. LAI AP)” column. Please revised the column, too. |
The medications prescribed are shown in Table 1b. “Other treatments” column has been changed. |
|
|
T The format of table 2 should be revised, too. The clinical characteristics (separated in table 1 and 2) in the oral antipsychotics and LAI group should be summarized in one table. The clinical characteristics of the standard and high dose group could be summarized in one table, too. |
The format of table 2 has been changed. The outcomes of all subgroups have been summarized in one table (Table 2). |
|
|
It would be better if the authors could add survival analysis and a figure regarding treatment discontinuation between oral and LAI treatment group. |
We have added a figure regarding treatment discontinuation through the follow-up (Figure 2). |
|
|
The hospitalizations in two group could also be presented as figures to show potential difference along the time course. |
The treatment abandons, hospitalizations and suicide attempts of all subgroups are presented as figures (Figure 1) |
|
|
P Please provide the definition of standard and high dose group in the methods. |
Definitions are provided in Methods. |
|
|
Please provide more details of reported side effects and laboratory test alteration in table 3 or in the results. |
More details of reported side effects and laboratory test alterations are reported in Table 3. |
|
|
I In addition to the CGI-S, was there any other measurements to evaluate the severity of psychotic symptoms in schizophrenia patients, such as PANSS? |
Unfortunately, any other measurements to evaluate the severity such as PANSS were not performed in all patients. |
|
Round 2
Reviewer 1 Report
The changes made by the authors are significant and largely correspond to the comments of the reviewers. However, I have one remark regarding tables: 1a and 2. In the case of statistical significance, we do not know which comparisons they refer to - that is, between which groups the comparisons turned out to be significant. I believe that this requires clarification and the text should better correspond and more clearly present the results contained in the above-mentioned tables.
Author Response
First of all, thank you for the comments and remarks.
Regarding tables 1a and 2, we have clarified, in the case of statistical significance, between which groups the comparisons turned out to be significant.
And, in the text, we have more clearly presented the results contained in the above-mentioned tables.